# Tool Condition Monitoring Using Machine Tool Spindle Current and Long Short-Term Memory Neural Network Model Analysis

**DOI:** 10.3390/s24082490

**Published:** 2024-04-12

**Authors:** Niko Turšič, Simon Klančnik

**Affiliations:** Faculty of Mechanical Engineering, University of Maribor, Smetanova ul. 17, 2000 Maribor, Slovenia

**Keywords:** tool condition monitoring, artificial intelligence, LSTM neural network

## Abstract

In cutting processes, tool condition affects the quality of the manufactured parts. As such, an essential component to prevent unplanned downtime and to assure machining quality is having information about the state of the cutting tool. The primary function of it is to alert the operator that the tool has reached or is reaching a level of wear beyond which behaviour is unreliable. In this paper, the tool condition is being monitored by analysing the electric current on the main spindle via an artificial intelligence model utilising an LSTM neural network. In the current study, the tool is monitored while working on a cylindrical raw piece made of AA6013 aluminium alloy with a custom polycrystalline diamond tool for the purposes of monitoring the wear of these tools. Spindle current characteristics were obtained using external measuring equipment to not influence the operation of the machine included in a larger production line. As a novel approach, an artificial intelligence model based on an LSTM neural network is utilised for the analysis of the spindle current obtained during a manufacturing cycle and assessing the tool wear range in real time. The neural network was designed and trained to notice significant characteristics of the captured current signal. The conducted research serves as a proof of concept for the use of an LSTM neural network-based model as a method of monitoring the condition of cutting tools.

## 1. Introduction

The state of the cutting tool is a vital part of reaching the quality targets for machined components. As such, the lack of this information leads to uncertainty in manufacturing and has various consequences for the process as a whole. These consequences include a drop in product quality, unplanned downtime causing losses in productivity, and, of course, all the economic losses associated. During the machining of aluminium alloys, a turning tool experiences different mechanisms of wear; the primary mechanism of wear is adhesion, followed by diffusion at elevated temperatures, as well as oxidative wear [1]. The unplanned downtime that occurs as a direct consequence of excessive tool wear was estimated to be 7 to 20 percent [2].

For Industry 4.0, a major component is prognostic and health management systems (PHM) [3] that focus on efficiently detecting industrial components that have deviated from normal operation parameters or predicting when failures are likely to occur.

Thus, a reliable, real-time tool condition monitoring system is the heart of intelligent manufacturing and autonomous production lines [4]. There are two categories of tool condition monitoring systems, i.e., direct and indirect methods. With direct methods, the state of the cutting tool is evaluated visually through an optical microscope. The method has a high degree of accuracy and gives the best evaluation of the tool condition. It has, however, serious real-time limitations as it requires the cutting process to be interrupted to analyse the tool. Further, it requires specific optical equipment, which is not appropriate for the industrial environment in which these systems are to be installed. The indirect methods, despite providing lower accuracy, are much simpler to implement in industrial applications for economic reasons and their resistance to contaminated environments. They rely on correlating one or more sensor signals to the tool wear state. For this, multiple variables and detection methods can be used [5].

The cutting force signal is one of the most reliable, as it is in direct correlation to the state of the cutting edge and is therefore very sensitive to changes in tool state [5,6]. In their research, Li et al. [6] introduced a force-based system, with experimental results reaching 96.76% accuracy. Research by leading authors, such as the work carried out by Amigo et al. [7] and Urbikain et al. [8], focused on cutting force signals in relation to tool wear, has confirmed that in terms of pure accuracy, this is one of, if not the best, indirect measure signal sources. Despite their excellent qualities, the use of force-based systems has some drawbacks, as obtaining these values requires sensitive dynamometers and often other sensors. The use of these types of sensors in an industrial setting has problems of practicality due to environmental contamination as well as cost effectiveness.

Another interesting approach is the utilisation of thermography for tool wear analysis. Brili et al. [9] utilised the images captured by a thermal imagery camera, extracting features that correlate with the tool state. The accuracy of the developed system ranged from 96.25% to 100%. Despite the excellent accuracy of the developed system, it has similar problems as the cutting force approach, mainly in terms of cost effectiveness when implementing it in an industrial setting.

Systems for industrial applications gravitate towards low-cost, robust sensors that do not interfere with the monitored process. Rmili et al. [10] developed a system that correlates the vibrations that are produced during machining to tool wear. The proposed method is interesting; however, it has certain limitations. Specifically, the vibratory signals are very sensitive and influenced by many variables (environment, location of sensors, type of cutting fluid, etc.), a lot of which cannot be kept consistent for an industrial implementation.

For an industrial implementation of a system, it is always the most practical to use the machine’s internal data, such as power or electric current. This allows for even easier integration and is also the most cost-effective. As such, a lot of research has been conducted in the area of utilising these signals. Drouillet et al. [2] used the spindle internal power sensor signal to predict the remaining useful life of the tool (RUL). Jamshidi et al.’s [4] research focused on utilising the current signal to develop an alerting system that sends a warning before the tool wear begins to reach a critical level. Our research focuses on these same signals, as the overview of the literature showed that it is a highly accessible signal that requires no additional modifications or additions to the machine, with only a slight decrease in the tool wear to signal correlation.

There exist different methods of signal analysis for the purposes of TCMs. Among those, we can include conventional statistical analysis [11], the combination of time and frequency analysis [12], fractal analysis [4], as well as utilising artificial intelligence, such as genetic algorithms [13], and image analysis with a convolutional neural network [9]. There are also frameworks that further improve convolutional neural networks. One such framework is spatiotemporal pattern networks (STPN), which enable adaptive feature learning when dealing with multivariate time series. Together, these networks are referred to as ST-CNN [14].

Along with the type of signal and the analysis method, a third vital part of a tool for condition monitoring systems is data processing. As raw data can rarely be used without problems, additional processing is required to extract relevant features from the data. In their research, Tapia et al. utilised an interquartile range [15], whereas Aldekoa et al. divided their data into areas of interest based on their knowledge of the manufacturing process [16].

The nature of our observed signals and their end application supports the use of an artificial intelligence approach. Particularly promising for data series analysis are recurrent neural networks such as long short-term memory networks [17,18]. These networks provide the ability to shift the focus of the analysis from general statistical parameters and absolute values, allowing for the recognition of inherent patterns in the acquired signals [17].

The tool condition monitoring concept proposed also looks to add redundancy to the LSTM network predictions. This addition, in the form of a unanimous voting algorithm (UVA), works to remove spontaneous errors and even further increases the classification accuracy of the TCM system.

In this study, an external measurement system was used to obtain the current data from the main spindle of a purpose designed CNC lathe. The reason for this style of measurement was to test the concept without altering the production line. The conducted research serves as a proof-of-concept for the use of an LSTM neural network model in combination with an UVA redundancy algorithm as a tool for condition monitoring in an industrial setting. The innovative approach allows for real-time tool monitoring with the potential for complete integration into the machine’s system, as shown in Figure 1.

This study intends to demonstrate that combining LSTM network models with existing voting-based concepts to add redundancy makes a classification system more reliable than existing models. Functionality is compared against multiple alternative models using more advanced sensing methods that do not use such support systems as the UVA used in the presented case.

The study has a strong focus on developing a system that is feasible for industrial implementation; thus, the model is based solely on the spindle current and no other signals, which allows for its introduction to be simple and without the need for large investments.

## 2. Materials and Methods

The main objective of this research is to prove that the current signal recorded on the main spindle is suitable and sufficient for determining the level of tool wear when analysed with a neural network classification model.

There is a considerable presence of noise in the current signal on the main spindle. That is the advantage of the recurrent neural network approach, as it allows for the analysis of not only the absolute current values but also the entire current graphed over the entire machining cycle.

The work is divided into the following: (I) experiment—acquisition of current time sequences during the milling cycle; (II) training a LSTM classification model—sequences filtered and divided into test and training groups, training the neural network on the latter group; (III) testing the model—the trained model was tested on previously unseen current signals; and (IV) improving functionality with an unanimous voting system algorithm (UVS)—improving the functionality by adding redundancy further improves the model’s accuracy.

### 2.1. Experiment

The turning process is affected by several factors as a consequence of machine properties, tool and mount type, cutting parameters, and external disturbances [6]. Despite obtaining the data from a production line in normal operation, most of these factors were held constant while the measurements were conducted. That is due to the CNC lathe being set up to conduct a single type of machining on consistent input material.

The current signal was recorded throughout the entire useful lifetime of the tool, which was roughly 1000 manufacturing cycles.

This style of data acquisition was conducted due to several factors, the chief of which was obtaining the data during normal operation of the production line. Due to limitations in obtaining data on the physical side, we decided to focus on data processing to eliminate unrepresentative current sequences (expected down times, outliers, etc.).

### 2.2. Categorising Tool Wear Levels

Tool wear can be analysed in several ways. Direct means, such as inspection of the tools under an optical microscope by an experienced examiner, can determine the wear level in great detail. There are also empirical methods, such as the Niakis method [19], that allow us to remove the human factor from the inspection process.

The main aim of the research is the development of an intelligent system that determines tool wear for the purposes of classifying tool wear and alerting the operator. For that reason, despite the fact that exact wear level prediction was feasible, a more practical approach was taken by assigning the tool wear to predetermined classes based on the actions required to maintain the production line. The wear classes were determined based on previous experience and the required quality standards of the final product. Currently, the tools are replaced every 1000 cycles to ensure sufficient surface quality. This led to separating the tool’s useful life into 3 categories (low, moderate, and high wear level).

As there was no secondary method to determine tool wear, buffer zones (50 machining cycles) were left between classes to protect against border cases in the training of the neural network. The breakdown of classes can be seen in Table 1.

### 2.3. Current Signal Acquisition

The current signal was recorded with external measurement equipment, and the effective current value was monitored at the output of the spindle drive (Sinamics CU320-2 PN), as well as other electrical parameters. The measuring equipment used was an MI2792 Power Q4 power quality analyser manufactured by Metrel. The main parameters of the device are provided in Table 2.

The measurements of the signal were conducted in 100 ms intervals and outputted as the average values of the effective current in those intervals. This signal-capturing method was used to reduce the amount of noise in the recorded dataset.

### 2.4. Experimental Setup

As mentioned, most of the machining parameters for the process we are obtaining our data from are already known (Table 3). These parameters are held constant and ensure that all the data obtained are acquired under identical circumstances, with the only significantly varying factor being tool wear. Ensuring the model is trained on a consistent database is paramount to generating a robust model.

The exact equipment used in the experiment and the cutting tool assembly are presented in Figure 2.

### 2.5. Long Short-Term Memory Network

The current-time sequences were classified using an artificial intelligence method. Specifically, a recurrent long short-term memory network. This style of artificial neural network was used because of its advantages when compared to regular feed-forward neural networks. In a classical neural network, the inputs are individual data points. This style of inputting data is applicable when different variables are used simultaneously to predict tool wear.

Typically, 3 to 8 input variables are used [20]. Examples include the following: number of revolutions, machining time, and cutting force [21]; material of the tool, the sharpening mode, the nominal diameter, the number of revolutions, the feed rate, and the drilling length [21]; depth of cut, cutting speed, and feed to the tooth [22]. The maximum number of inputs that were utilized successfully when using fully connected neural networks was 20 [23]. In the case of using time sequences as neural network inputs, such as in this study, a fully connected neural network would have more than 100 individual inputs, each being a separate datapoint.

Such an input net size would be too expansive for a fully connected layer to process, as it would create an expansive network with a high training and execution time. Not only that, but in this method, the seemingly significant parts of the current signal would have to be determined to reduce complexity, adding to the risk of potentially faulty personal perception.

By using an LSTM memory network, these issues can be avoided, as recursive neural networks can take data sequences as single inputs. This type of network also does not require the significant parts of the signal to be determined in advance, as the network is trained to analyse not only the individual data points but also the relationships between them. This results in the network analysing the shape of the signal rather than the absolute current values.

What differentiates this type of neural network from a classic feed-forward network is the recurrent LSTM layer consisting of repeating modules, as shown in Figure 3. These contain 4 independent neural networks that allow the network to add or remove information in the training phase.

Mathematically, this layer is described with the following formulae [24]:(1)$$f_{t}=\sigma (W_{f}*\left[h_{t-1\;},\;x_{t}\right]+b_{f}$$
(2)$$i_{t}=\sigma (W_{i}*\left[h_{t-1\;},\;x_{t}\right]+b_{i}$$
(3)$$o_{t}=\sigma (W_{o}*\left[h_{t-1\;},\;x_{t}\right]+b_{o}$$
(4)$${\tilde{c}}_{t}=tanh(W_{c}*\left[h_{t-1\;},\;x_{t}\right]+b_{c}$$
(5)$$c_{t}=f_{t}⊙c_{t-1}+i_{t}⊙{\tilde{c}}_{t}\;$$
(6)$$h_{t}=o_{t}⊙\tan h\left(c_{t}\right)$$
where the initial values are *c_o_* = 0 and *h_o_* = 0, and the operator ⊙ denotes an element-wise product. Where *d* and *h* refer to the number of input sequences and the number of hidden units, respectively, the variables are defined as:
xt∈ℝd: input vector to the LSTM module;ft∈(0,1)h: activation vector of the forget gate;it∈(0,1)h: update gate activation vector;ot∈(0,1)h: output gate activation vector;ht∈(−1,1)h: hidden state or LSTM output vector;c˜t∈ℝh: cell input activation vector;ct∈ℝh: cell value vector;W∈ℝhxd and b∈ℝh: matrices of NN weights and bias vector parameters that adapt during training.

The development of LSTM is a lengthy process; however, starting point structures are freely available. In the study, some of the network’s parameters were additionally optimised to achieve the best compromise between functionality and response time. The parameters chosen for optimisation were the number of training iterations and the network learning rate (see Figure 4 and Figure 5).

### 2.6. Unanimous Voting System Algorithm

To additionally reduce the occurrence of false alarms in the final monitoring application, a way to add redundancy in the analysis was required. The proposed solution is a simple voting system where the neural network model triggers the appropriate alarm based on multiple consecutive analyses, requiring them to be identical.

The unanimous voting system is a basic algorithm that compares 3 consecutive classifications that the neural network prediction model performs in real time. When the system recognises an identical wear level after three machining cycles in a row, the process can be interrupted, with the minimum risk of a false alarm downtime. The flowchart for the final monitoring application can be seen in Figure 6.

## 3. Results and Discussion

In Section 2.3, the process of acquiring the spindle current data in relation to time as well as the distribution of that data into three categories of tool wear is presented. The entire data set was split into two groups for training and testing the neural network. Initially, the LSTMNN was trained on a set of designated training data, and later, the performance of the model was further tested with a set of testing signals—ones the network did not encounter during training. The size and distribution of the described sets are given in Table 4.

The classification quality was evaluated using the parameters of accuracy, recall, and precision [25] to ensure reliability. To visualise the results of the model’s classification attempts, contingency tables were used.

The trained model was then implemented with the unanimous voting algorithm and had its accuracy tested again.

### 3.1. Training Set Results

Firstly, the training process was conducted, and then all the training data were classified. The resulting contingency table is depicted in Figure 7.

A brisk inspection of Figure 7 shows that a trained network correctly determined 1102 out of 1106 current signals after being tasked with classifying them.

The signals that resulted in false predictions were additionally analysed to determine the error source. Inspection showed that the false prediction resulted from current data sequences that were located on the edges of the wear intervals (see Table 1). Thus, the false predictions are mostly concentrated in the “moderate wear” class with two border zones.

### 3.2. Testing Set Results

Firstly, the training process was conducted, and then all the training data were classified. The resulting contingency table is depicted in Figure 8.

The change in model performance can be most easily evaluated via accuracy. A slight decrease in the said parameter can be observed between the training and testing data classifications (99.64% falling down to 96.93%), but not outside of what is expected.

The network correctly categorised 347 out of 358 current signals. When observing the error distribution, the bulk of the models’ errors are in the “moderate wear” class, as was observed with the categorisation of training data. Additionally, all but one of the errors are one class over, with a tendency towards the higher wear class. This fact is important from a practical point of view for the intended implementation of such a system in an industrial setting. A random error distribution, despite its high accuracy, would be unfavourable.

### 3.3. Unanimous Voting System (UVS)

While a classification accuracy of ~97% is certainly high, additional methods were consulted to improve the reliability of the model even further. A method that was implemented is an inherent redundancy that requires the model to make three consecutive identical tool wear estimations (Figure 8), before resulting in an alarm or notification to the process overview interface. A more detailed description can be found in Section 2.6.

This, along with the inherent model accuracy of 96.93%, increases the predictive system’s accuracy to effectively 100% (99.99%).

## 4. Discussion

The study presented addresses the following two main questions: whether the condition of the cutting tool used in CNC turning can be monitored with an artificial neural network, and whether the current signal captured on the main spindle is a sufficient feature for that application.

The current sequences are analysed using a long short-term memory recurrent neural network. An individual current signal data sequence consisted of a continuous 5 s spindle current measurement coinciding with the machining cycle. The obtained sequences of data were then divided into groups based on the age of the cutting tool in terms of machining cycles. The conclusions of the research are the following:Cutting tools were successfully divided into groups according to tool wear.The proposed method is confirmed to be applicable to a tool condition monitoring system.The current signal is a sufficient feature for determining tool wear.The accuracy of the proposed system ranges from 96.93% to effectively 100%, with additional wear prediction redundancy in the form of a UVA.

The results are more than encouraging when compared to other studies utilising the neural network modelling approach. The results are as follows: 5%, 10.7%, and 22% errors for estimated tool wear for milling tools [26]; 99% accuracy for useful life prediction using a neuro-fuzzy network with wireless sensor [27]; comparison of predicting exact tool wear with different algorithms, where the LSTM reached an accuracy of 92.54% [28]; a similar study classifying the tool wear rather than predicting the exact value [29] achieved an accuracy of 95.25%, the highest when compared to RNN (85.26%) and Feedforward NN (79.35%). Convolutional neural networks are also commonly used, reaching accuracy as high as 96.25% [9].

In Table 5, the methods are further described and compared to the results of our research. The main comparisons that are being focused on are the overall accuracy of the models as well as the measuring equipment required to obtain the signal on which the predictions are based. The latter is used to illustrate the applicability of such a system for an industrial application, where low cost and resistance to contamination are prioritized.

The proposed classification system based on LSTM Neural Network spindle current analysis with added redundancy is more accurate and allows for easier integration than other similar TCM systems.

## 5. Conclusions and Future Work

The system presented is an intelligent tool condition monitoring solution, showing high potential with its high accuracy and reliability, low investment cost (most necessary adaptations are digital), and high level of possible integration into the process machinery. It serves the goal of removing unpredictable decision-making based on human operator experience and knowledge, in line with the ideals of Industry 4.0.

The future development of this system will include improving data acquisition, expanding the dataset, and using more reliable measures of establishing tool wear in the training data. The latter can be achieved by visually inspecting the tool plates or utilizing the Niakis method [19]. Further, it is intended to expand upon the way class distribution is determined and increase the number of said classes. These would include classes to address the areas of uncertainty that can be observed when classifying the cases on the borders of wear intervals.

The generalizability of the model has not been directly examined in this study. However, based on other studies in the field of machine learning-based TCM systems [31], it can be concluded that LSTM networks have the ability to reach a certain level of generalisation, especially when utilised with existing redundancy concepts. This is also one of the future development goals for the system. By training and testing on a larger dataset with variations in the machining parameters such as cutting speed, feed rate, and cut depth, as well as changes in material, we could develop a robust system that can adjust to changes in manufacturing conditions. Additionally, more advanced methods of integrating redundancy would be considered, such as majority voting utilising multiple LSTM models trained on similar but separate data.

Finally, as industrial applicability is a large goal, utilising the machine’s internal sensors to record the current signal is a must. The ability to use the signals already existing on the equipment would be a great benefit, both in terms of reducing cost and complexity.

## Figures and Tables

**Figure 1 sensors-24-02490-f001:**
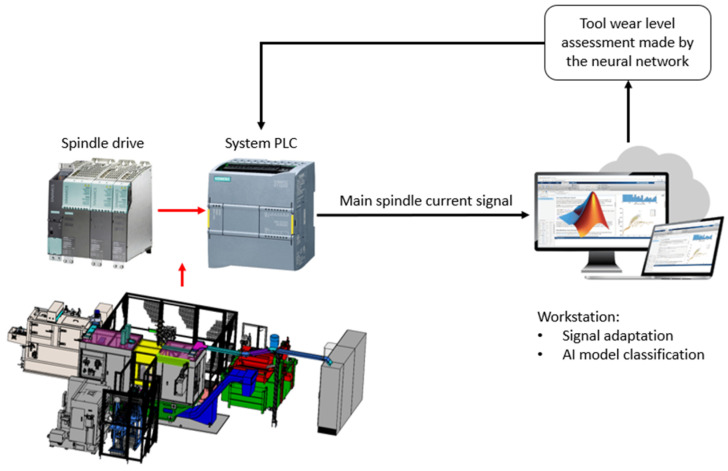
A schematic overview of a proposed tool condition monitoring system using AI to assess tool wear based on the spindle electric current signal.

**Figure 2 sensors-24-02490-f002:**
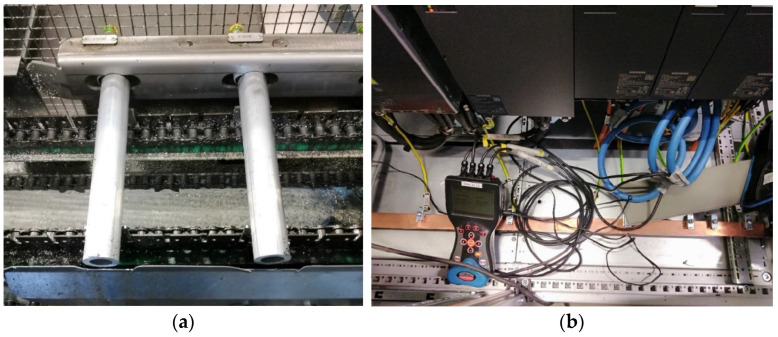
Key components of the experiment include: (**a**) workpieces utilised in the machining process; (**b**) configuration of the power quality analyser to assess the output from the spindle drive; (**c**) assembly of the cutting tool; and (**d**) geometry of the cutting plate.

**Figure 3 sensors-24-02490-f003:**
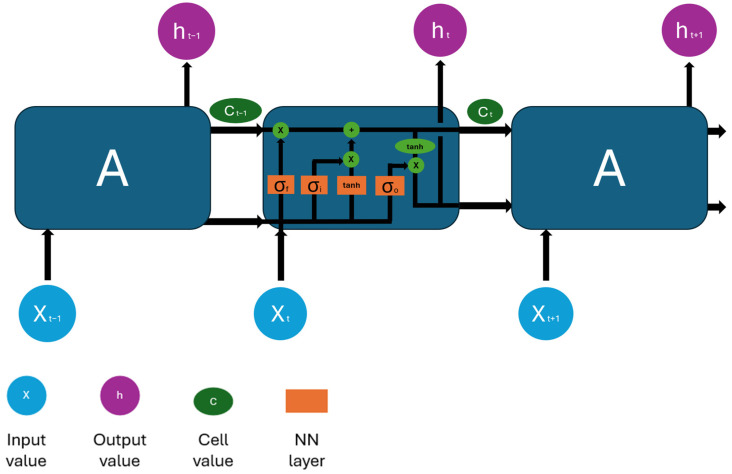
Schematic representation of the LSTM recurring layer.

**Figure 4 sensors-24-02490-f004:**
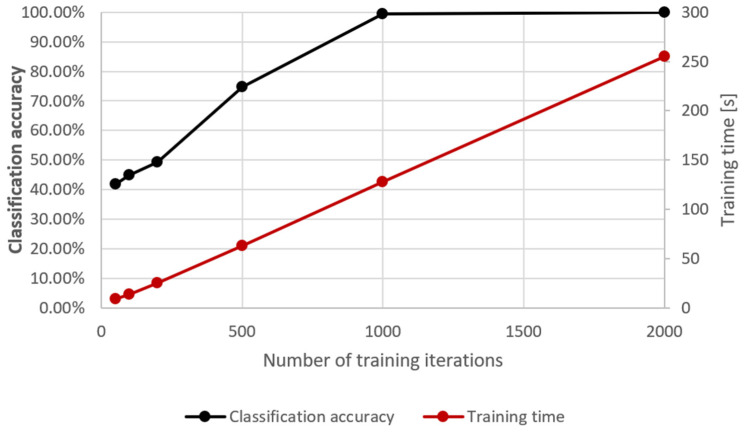
Optimisation of the number of training iterations against classification accuracy and the training cycle duration.

**Figure 5 sensors-24-02490-f005:**
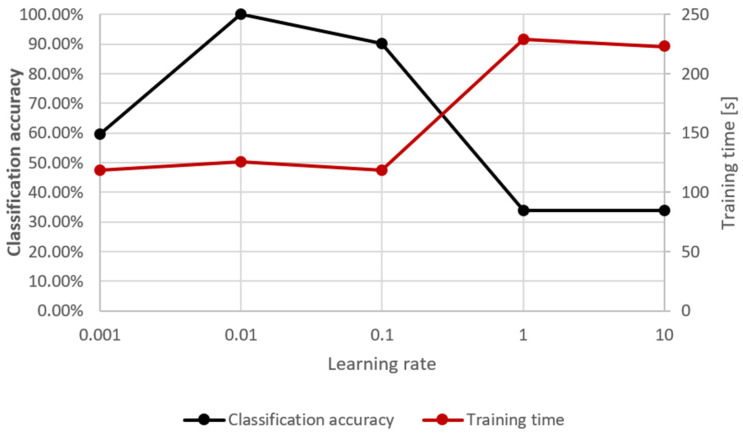
Optimisation of the learning rate against classification accuracy and the training cycle duration.

**Figure 6 sensors-24-02490-f006:**
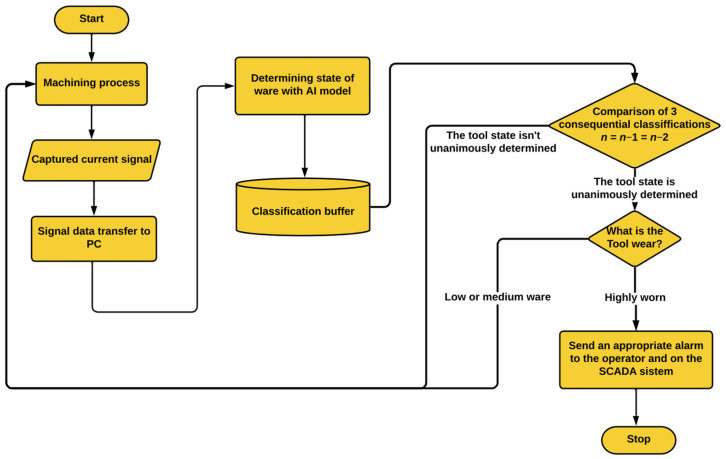
The flowchart for the final tool condition monitoring application that has an integrated algorithm for additional redundancy.

**Figure 7 sensors-24-02490-f007:**
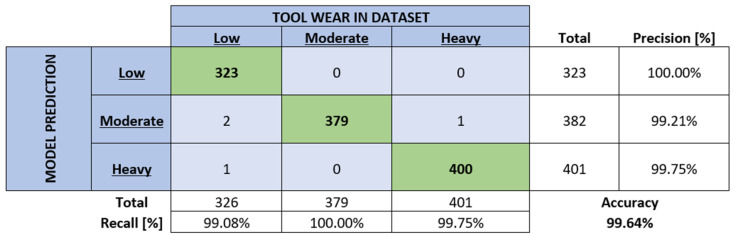
Contingency table of classification results when analysing the training data set.

**Figure 8 sensors-24-02490-f008:**
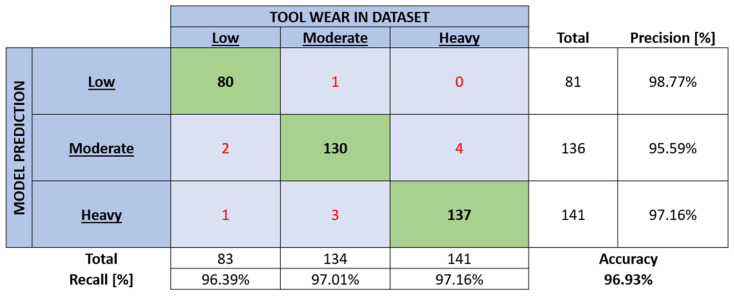
Contingency table of classification results when analysing the testing data set.

**Table 1 sensors-24-02490-t001:** Tool wear class assignment based on the number of completed machining cycles.

Number of Machining Cycles	Class
0–300	Low wear
350–650	Moderate wear
700–1000	High wear

**Table 2 sensors-24-02490-t002:** Specifications of the power quality analyser MI2792 Power Q4.

Measurement Method	Measuring Range	Measurement Accuracy	Response Time
Flexible current clamps	1 mA–30 A	±0.25%	<1 µs

**Table 3 sensors-24-02490-t003:** Boundary conditions and constant parameters in our experimental setup.

Part of Experimental Setup	Parameter	Basic Experiment
Cutting parameters	Cutting depth	0.5 mm
	Cutting speed	530 m/min
	Feed rate	3.3 m/min
	Cooling	External emulsion cooling
Workpiece material	Material type	Aluminium alloy
	Producer	Impol LLT
	EN	AW-6082
	Yield strength	260 MPa
Cutting tool	Type	Custom designed
	Producer	Walter tools
	Material	Polycrystalline diamond
	Geometry	Positive
	Corner radius	1.2 mm
	Cutting edge length	12 mm
Lathe	Producer	Unior
	Type	572-0000-0

**Table 4 sensors-24-02490-t004:** Number of current data sequences in the training and testing data sets.

Element	Low	Moderate	High	Combined
Training set	326	379	401	1106
Testing set	81	136	141	358

**Table 5 sensors-24-02490-t005:** Comparison of the model proposed (first line) in the study against existing research.

Method	Observed Signal	Measuring Requirements	Accuracy	Reference
LSTM neural network	Current	Internal current sensors	96.93–99.99%	/
Modal analysis	Vibrations	Dynamometers	78–95%	[26]
Neuro-fuzzy network	Vibrations	Wireless accelerometer	99%	[27]
LSTM neural network	Force	Multiple external sensors	92.54%	[28]
LSTM-hidden Markov model	Force	Dynamometer	95.25%	[29]
CNN	Thermal image	Thermal camera	96.25%	[9]
CNN	Acoustic signal	Spherical beamformer	65–99.5%	[30]

## Data Availability

Data Availability Statements are available in all Sections.

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
