# Peer review of "Tool Condition Monitoring Using Machine Tool Spindle Current and Long Short-Term Memory Neural Network Model Analysis"

_sensors, 2024, doi:10.3390/s24082490_

Round 1

Reviewer 1 Report

Comments and Suggestions for Authors

This paper discusses a tool monitoring captability using an LSTM model. Although interesting, I believe this paper is in need of heavy revision prior to acceptance. I recommend, rejection.

1. Abstract should be one cohesive paragraph rather than small excerpts of 3 sentences. Revise.

2. Introduction is far too general and short. The authors only included 20 citations total, which is not sufficient for a journal article. Please add a significant amount of references that outline the current state of the art.

3. The authors claim that they will be using an LSTM for label classification.  I am struggling to see the scientific novelty of this paper. This has been widely accepted in the field.

4. All figure captions are in need of revision. They are not informative whatsoever.

5. Figure 5 is hard to read and digest. Regenerate entirely.

6. The confusion matricies are not properly formatted. Commas should be replaced with a decimal point (i.e., 99.9% rather than 99,99%).

7. This paper is lacking flow and a sound description of the theoretical contributions. There is limited explanation of the mathematics that go behind this work. It reads to me more like an extended abstract rather than a journal article.

Comments on the Quality of English Language

Adequate, but in need of minor revision.

Author Response

1. Abstract should be one cohesive paragraph rather than small excerpts of 3 sentences. Revise.

Abstract was reformatted into one cohesive paragraph.

2. Introduction is far too general and short. The authors only included 20 citations total, which is not sufficient for a journal article. Please add a significant number of references that outline the current state of the art.

The existing indroduction has been expanded upon with a more detailed overview of the current state of the art and other concepts in the TCM area of research.

Additional discussion was added to further compare our results with existing research.

3. The authors claim that they will be using an LSTM for label classification. I am struggling to see the scientific novelty of this paper. This has been widely accepted in the field.

Additional content was added expanding upon the novelty of this research.

The focus is not solely on the LSTM network's ability to classify labels, but also building upon existing systems by adding traditional redundancy concepts for higher applicability in an industrial setting. Thus, making models using simpler to acquire data on par with more sophisticated methods.

4. All figure captions are in need of revision. They are not informative whatsoever.

Additional information has been added to the figure descriptions.

5. Figure 5 is hard to read and digest. Regenerate entirely.

New figures have been provided. The graphs were separated into 2 figures for better clarity. The previously included figure was added by mistake.

6. The confusion matricies are not properly formatted. Commas should be replaced with a decimal point (i.e., 99.9% rather than 99,99%).

The improper formatting was corrected.

7. This paper is lacking flow and a sound description of the theoretical contributions. There is limited explanation of the mathematics that go behind this work. It reads to me more like an extended abstract rather than a journal article.

Additional explanation of the mathematical/theoretical concepts was added in chapter 2. Materials and methods, better explaining LSTM neural networks.

Reviewer 2 Report

Comments and Suggestions for Authors

The manuscript presents a method for tool condition monitoring using a LSTM neural network to analyze spindle current in machining processes. Despite the relevance of the work to Industry 4.0 initiatives, the contribution is diminished by a lack of demonstrated novelty, inadequate comparison with existing methodologies, and failure to address domain shift issues and the IID data assumption. These gaps significantly undermine the manuscript's impact and applicability in the broader context of manufacturing technologies.

1. The paper claims the application of an LSTM neural network for TCM as novel. However, the utilization of LSTM for monitoring various conditions, including tool wear, has been explored previously. The manuscript needs to clearly articulate the novel aspects of the proposed approach compared to existing solutions. A more thorough literature review is necessary to position the work within the current state of the art.

2. The manuscript does not adequately compare the proposed model's performance with other existing methods or models in the field of TCM. It is important to benchmark the LSTM model against other AI-based or conventional methods to highlight its advantages or improvements in accuracy, reliability, or cost-efficiency.

3. The study does not consider the problem of domain shift, which is crucial for the generalizability of the model to different manufacturing environments or materials. The ability of the model to adapt to variations in machining conditions, such as different materials or cutting parameters, should be evaluated to ensure its industrial applicability.

4. While the manuscript mentions future developments regarding data acquisition and enhancing model robustness, it would be beneficial to provide preliminary ideas or strategies on how these aspects will be tackled. Expanding the dataset and incorporating machine-internal sensors are positive steps, but the approach for dealing with variations in machining parameters and expanding the diagnosis parameters should be more detailed.

5. Please provide a more thorough literature review on the topic of PHM, including recent advances in data-driven methods and their limitations in practical engineering applications. This will help readers understand the significance and novelty of your proposed method. Other relevant methods should be included in the Lit Review section, such as the following papers: 10.1109/TIM.2023.3259048, doi.org/10.1016/j.ymssp.2018.07.048.

Comments on the Quality of English Language

Moderate editing of English language required.

Author Response

1. The paper claims the application of an LSTM neural network for TCM as novel. However, the utilization of LSTM for monitoring various conditions, including tool wear, has been explored previously. The manuscript needs to clearly articulate the novel aspects of the proposed approach compared to existing solutions. A more thorough literature review is necessary to position the work within the current state of the art.

Additional content was added expanding upon the novelty of this research.

The focus is not solely on the LSTM network's ability to classify labels, but also building upon existing systems by adding traditional redundancy concepts for higher applicability in an industrial setting. Thus, making models using simpler to acquire data on par with more sophisticated methods.

2. The manuscript does not adequately compare the proposed model's performance with other existing methods or models in the field of TCM. It is important to benchmark the LSTM model against other AI-based or conventional methods to highlight its advantages or improvements in accuracy, reliability, or cost-efficiency.

Additional comparisons have been provided in the discussion portion of the article, comparing the model to other AI based systems with varying indirect measurement signals.

3. The study does not consider the problem of domain shift, which is crucial for the generalizability of the model to different manufacturing environments or materials. The ability of the model to adapt to variations in machining conditions, such as different materials or cutting parameters, should be evaluated to ensure its industrial applicability.

Discussion of domain shift has been added to the article.

The generalizability of the model is not directly discussed, but based on other studies in the field, we can conclude that LSTM networks have the ability to generalize. In order to achieve that with our model further research with a larger data base is required as suggested.   

4. While the manuscript mentions future developments regarding data acquisition and enhancing model robustness, it would be beneficial to provide preliminary ideas or strategies on how these aspects will be tackled. Expanding the dataset and incorporating machine-internal sensors are positive steps, but the approach for dealing with variations in machining parameters and expanding the diagnosis parameters should be more detailed.

This was further expanded upon, based on your review.

5. Please provide a more thorough literature review on the topic of PHM, including recent advances in data-driven methods and their limitations in practical engineering applications. This will help readers understand the significance and novelty of your proposed method. Other relevant methods should be included in the Lit Review section, such as the following papers: 10.1109/TIM.2023.3259048, doi.org/10.1016/j.ymssp.2018.07.048.

A further look at the current state of technology including PHM datasets has been added to the article.

Reviewer 3 Report

Comments and Suggestions for Authors

my comments are in the file attached.

Author Response

1. Monitoring is key but two things are really important: the real significance of data in the parameter monitored (see Implementation of a scalable platform for real-time monitoring of machine tools, Computers in Industry 155, 104065) and the machine learning approach used, see Mechanical Systems and Signal Processing 204, 110773 and both aspect need to ne enhance through the paper. Indirect measures and virtual sensors are a trend now.

The additional overview of methods to determine data significance has been added to the article.

2. The indirect methods, despite providing a lower accuracy are much simpler and robust: (line 46) this is very under discussion, please eliminate to positive arguments. Direct are much better.

Upon further review we decided to revise this argument and further work on the section elaborating our reasoning for using indirect methods.

3. Cutting force signal: many people studied that as it was in several papers, those related with https://doi.org/10.1016/j.aej.2022.10.060 toolholder are key but there were others, see recent Works by leading authors: Amigo, Urbikain, etc.

Upon your suggestion we have looked into the work of the suggested authors and revised the literature overview accordingly.

4. Current is one indirect method, people used Hall effect sensors many times. Indirect can be compared and some authors did some ideas in TCM system in contour milling of very thick-very large steel plates based on vibration and AE signals, in Journal of Materials Processing Technology.

In the discussion portion of the article, we added a more thorough comparison of our proposed model against other models based on different indirect methods, including vibrations, forces and acoustic signals, as well as others.

5. Table 1. Classes of tool ware based. What is ware?

 This was a simple typo, the tool “ware” has been correctly changed to “wear”.

6. Figures 2,3,4 please make better, include in only one.

Additional information was provided in figure description. The figures were also joined together as suggested (a, b, c and d).

8. Eliminate Table 4, the standard says the composition.

Removed as requested.

9. No images of tool wear? Why?

At the time of submitting the article certain materials were not provided by the company we were working with for our research. We have attempted to get these materials, but they did not want the pictures published.

10. Ref 1 and others are not related, ref 20 can be replaced by others.

Ref 1 has been replaced with one that better relates to the tool wear during machining of aluminium alloys.

We also replaced reference 20 with more current research.

Round 2

Reviewer 1 Report

Comments and Suggestions for Authors

I appreciate the authors willingness to adjust the manuscript, and I do believe it has added quality. 

Comments on the Quality of English Language

Adequate

Reviewer 2 Report

Comments and Suggestions for Authors

The authors seem to have addressed the review comments satisfactorily. The manuscript has been significantly improved. The reviewer would like to recommend the acceptance.

Comments on the Quality of English Language

The authors seem to have addressed the review comments satisfactorily. The manuscript has been significantly improved. The reviewer would like to recommend the acceptance.